# Oxidative Stress Correlates with More Aggressive Features in Thyroid Cancer

**DOI:** 10.3390/cancers14235857

**Published:** 2022-11-28

**Authors:** Marina Muzza, Gabriele Pogliaghi, Carla Colombo, Erika Carbone, Valentina Cirello, Sonia Palazzo, Francesco Frattini, Davide Gentilini, Giacomo Gazzano, Luca Persani, Laura Fugazzola

**Affiliations:** 1Endocrine Oncology Unit, Department of Endocrine and Metabolic Diseases, IRCCS Istituto Auxologico Italiano, 20095 Milan, Italy; 2Department of Pathophysiology and Transplantation, University of Milan, 20122 Milan, Italy; 3Pathology Unit, IRCCS Istituto Auxologico Italiano, 20145 Milan, Italy; 4Division of Surgery, IRCCS Istituto Auxologico Italiano, 20145 Milan, Italy; 5Bioinformatics and Statistical Genomics Unit, IRCCS Istituto Auxologico Italiano, 20145 Milan, Italy; 6Department of Brain and Behavioral Sciences, University of Pavia, 27100 Pavia, Italy; 7Department of Medical Biotechnology and Translational Medicine, University of Milan, 20122 Milan, Italy

**Keywords:** oxidative stress, H_2_O_2_, BRAF, thyroid cancer, thyroid adenoma

## Abstract

**Simple Summary:**

Several pieces of evidence indicate an impact of oxidative stress (OS) in the pathogenesis and progression of thyroid cancer. The aim of the study was to investigate OS in malignant and benign thyroid lesions and normal tissues, and to correlate the degree of OS with the genetic and expression profile of the tumors and clinico-pathologic features of the patients. The level of OS, considering all malignant and benign lesions, was higher than in normal tissues. In malignant tumors, OS was inversely correlated with tumor differentiation, and directly correlated with the presence of somatic mutations and with the worst tumor presentation and higher tumor aggressiveness. The increased OS status in thyroid tumors warrants further investigation since it may have diagnostic, prognostic and therapeutic relevance.

**Abstract:**

Oxidative stress (OS) can have an impact in the pathogenesis and in the progression of thyroid cancer. We investigated the levels of reactive oxygen species (ROS) in 50 malignant and benign thyroid lesions and 41 normal tissues, and correlated them with the thyroid differentiation score-TDS and the clinico-pathologic features. NOX4 expression, GPx activity and the genetic pattern of tumors were evaluated. In malignant and benign lesions, ROS generation and NOX4 protein expression were higher than in normal tissues. Follicular (FTCs) and anaplastic/poorly differentiated cancers had increased OS relative to papillary tumors (PTCs). Moreover, OS in FTCs was higher than in follicular adenomas. Mutated PTCs showed increased OS compared with non-mutated PTCs. In malignant tumors, OS was inversely correlated with TDS, and directly correlated with tumor stage and ATA risk. GPx activity was increased in tumors compared with normal tissues, and inversely correlated to OS. In conclusion, our data indicate that thyroid tumors are exposed to higher OS compared with normal tissues, while showing a compensative increased GPx activity. OS correlates with tumor aggressiveness and mutations in the MEK-ERK pathway in PTC. The inverse correlation between OS and TDS suggests that ROS may repress genes involved in thyroid differentiation.

## 1. Introduction

During the last decades an impressive rise in the incidence of thyroid cancer (TC) has been documented worldwide [1], though the survival curves are almost unchanged [2]. For differentiated thyroid cancer, radioiodine treatment (RAI) is extremely effective, but around 10% of cases can dedifferentiate, become RAI refractory and display an aggressive behavior [3]. 

In recent years, knowledge on the molecular alterations underlying thyroid cancer (TC) has progressively increased [4], unveiling new pathogenic mechanisms and opening promising targeted therapeutic opportunities. Among papillary thyroid cancers (PTCs), BRAF-like and RAS-like tumors have been identified, the former being associated with a decreased expression of several thyroid-specific genes, including sodium/iodide symporter (NIS), which may be responsible for the loss of RAI avidity [5]. Interestingly, genes involved in thyroid differentiation (on which expression the thyroid differentiation score is calculated) might also be silenced by a mechanism controlled by NADPH oxidase (NOX)-derived reactive oxygen species (ROS), suggesting a role of oxidative stress (OS) in RAI refractoriness [6]. 

OS is predicted to have a major impact both in the pathogenesis and in the progression of TC [7,8], since, due to the process of hormonogenesis, the thyroid gland is exposed to a high level of ROS [9]. Mild increases in ROS levels have been shown to induce oncogenes and to inhibit tumor suppressor expression and activity, while promoting the cross-talk between tumor cells and cancer-associated fibroblasts [7,10]. Beyond the electron transport chain of the mitochondria, the main source of cellular ROS is represented by NOXs. In the thyroid gland, three main NOXs have been identified: DUOX1 and 2, responsible for the calcium-dependent H_2_O_2_ generation used for iodide organification, and NOX4, which constitutively produces ROS in intracellular compartments [11]. An increased expression of both DUOX and NOX4 has been documented in TC [12]. Consistently, we previously detected a higher H_2_O_2_ production from either DUOX and NOX in PTCs compared with normal thyroid tissues [13].

Of note, NOX4 was found to be upregulated by BRAFV600E via a TGF-β-Smad3-dependent pathway in thyroid cancer cells and the level of NOX4 expression is increased in human and murine BRAFV600E-mutated thyroid tumors and inversely correlated with thyroid differentiation [7]. 

ROS detoxification occurs thanks to the action of some enzymes, such as catalase, and the glutathione peroxidase (Gpx)/glutathione reductase (GR) and peroxiredoxin (Prx)/thioredoxin reductase (TrxR) systems [14,15,16]. Among them, the selenium-containing GPx enzymes seem to play the major role in thyrocytes [17,18], and GPx1 isoform, found in cytosol, mitochondria and peroxisomes, is the most abundant [19]. Discordant data are available about its expression in thyroid tissues. In particular, GPx1 expression and GPx activity were found to be higher in human thyroid tumors compared with normal thyroid tissue (https://www.cancer.gov/about-nci/organization/ccg/research/structural-genomics/tcga, accessed on 30 September 2022; https://www.gtexportal.org/home/aboutGTEx, accessed on 30 September 2022), [13,20,21]. In contrast, other studies reported that GPx1 expression is reduced in thyroid tumors, and associated with an increased level of free radicals in the tumor tissue [22,23]. 

The aims of the study were to expand our previous findings on OS and GPx activity, which were obtained in a PTC series. In particular, we studied a new PTC series and a cohort of thyroid cancers with different histotypes, benign thyroid lesions and corresponding normal tissues. Moreover, data obtained were correlated with the genetic profile, the thyroid differentiation score (TDS) and the clinical and pathological data of patients. 

## 2. Materials and Methods

### 2.1. Cases 

Twenty-three PTCs (different from those included in Muzza et al., 2016 [13]), 6 FTCs, 4 PDTCs, 3 ATCs, 4 non-invasive follicular thyroid neoplasms with papillary-like nuclear features (NIFPTs), 7 follicular adenomas (FAs), 3 Hurtle adenomas (HAs) and 41 corresponding contralateral normal tissues were included in the study. All available specimens were reviewed by a senior pathologist to confirm the diagnosis. Tumors were classified and staged according to the 8th edition of the TNM staging (AJCC) [24]. All patients were included in a comprehensive database which contained several clinical and pathologic information. Criteria used to identify remission or persistent/recurrent disease were drawn on the bases of the American guidelines for the management of differentiated thyroid cancer [25]. The clinical features of the malignant tumors included are reported in Table 1.

### 2.2. Cellular Fractioning

Surgery samples were cut to remove fibrous tissue and homogenized using a Dounce homogenizer in 50 mM sodium phosphate buffer (pH 7.2) containing 0.25 M sucrose, 0.5 mM dithiothreitol, 1 mM EGTA and protease inhibitors (Roche). Homogenates were filtered, and centrifuged at 500× *g* at 4 °C to remove intact cells. The supernatant was initially centrifuged at 630× *g* for 10 min to isolate the nuclear fraction and then at 10,000× *g* for 20 min, the supernatant being collected as a cytoplasmic fraction. The protein concentration was measured by Pierce BCA assay (ThermoFisher, Waltham, MA, USA). The nuclear and the cytoplasmic fractions were stored at −80 °C for subsequent analyses.

### 2.3. OS Measurement in Tissues

OS was evaluated by measuring ROS production by NOXs in the cytoplasmic fraction of both benign and malignant tumors and normal tissues. In particular, 20 µg of cytoplasmic fraction were incubated in 150 mM sodium phosphate buffer (pH 7.4) containing 1 mM EGTA, 100 U/mL superoxide dismutase (SOD), 0.5 U/mL horseradish peroxidase (Merck, Rahway, NJ, USA) and 50 mM Amplex Red Reagent (Invitrogen, Waltham, MA, USA), and the fluorescence was measured in a microplate reader at 37 °C, using wavelength excitation at 535 nm and emission at 610 nm (Vitor2, PerkinElmer, Waltham, MA, USA). H_2_O_2_ production was quantified using standard calibration curves and specific activity was expressed as nmol per milligram of protein per hour. Each sample was analyzed at least in two independent experiments and results were expressed as averages. 

### 2.4. Genetic Characterization 

The molecular profile of tumor samples was characterized by a custom mass spectrometry PTC-MA panel (Sequenom, Agena, San Diego, CA, USA), as we previously reported [26,27]. In particular DNA and RNA were extracted from the nuclear fractions using an All Prep DNA/RNA kit (Qiagen, Hilden, Germany). RNA was reverse-transcribed (Reverse Transcription System, Promega, Madison, WI, USA) and three different assays were used to interrogate 15 point-mutations in 8 genes (BRAF, H-RAS, N-RAS, K-RAS, EIF1AX, AKT1, PIK3CA, TERT) in DNA and 7 gene fusions (RET-PTC1, 2 and 3, TRK, TRKT1, and T3 and PAX8-PPARγ) in cDNA. The molecular analysis of TP53 (exons 2–11) and PTEN (exons 5–8) was performed by PCR amplification and direct sequencing [28].

### 2.5. Expression Characterization and Thyroid Differentiation Score (TDS) Calculation

TDS was calculated in all PTCs, 4 FTCs, 2 NIFTPs, 2 FAs, 2 HAs, 4 PDTC/ATC and, as controls, in 2 pools, each obtained by mixing 20 normal thyroid tissues (10 from women and 10 from men). A custom AmpliSeq RNA expression panel (Illumina) was designed by the GenomeStudio software to detect the expression of 16 thyroid function genes (DIO1, DIO2, DUOX1, DUOX2, FOXE1, GLIS3, NKX21, PAX8, SLC26A4, SLC5A5, SLC5A8, TG, THRA, THRB, TSHR and TPO) and of the housekeeping gene TBP. mRNA libraries were constructed with an input of 20 ng of RNA using the AmpliSeq Library Plus (Illumina), according to the manufacturer’s recommendations. Constructed libraries were sequenced by MiSeq Reagent kit v.3 on an Illumina MiSeq sequencer. Basic data analysis was performed according to the default parameters of the Illumina’s MiSeq Reporter software. The mRNA expression level of the 16 thyroid function genes was used to derive the Thyroid Differentiation Score (TDS). In particular, reads were normalized through the DESeq2 R package [29] and transformed with log2(gene)/log2TBP. The TDS for each sample was obtained by extracting the mean of the 16 genes analyzed [4].

### 2.6. NOX4 Expression 

NOX4 protein expression in the cytoplasmic fraction was measured in normal and neoplastic tissues by Western blot in 12 PTCs, 2 NIFTPs, 2 FAs, 2 HAs, 5 FTCs, 4 PDTC/ATCs and 9 normal tissues. In particular, 30 μg of cytoplasmic fraction was resolved by electrophoresis on a gradient of 6–12% SDS-PAGE under reducing conditions and immunoblotted with a rabbit anti-NOX4 (Novus Biologicals) and a mouse anti-β-actin (Thermofisher). A loading control (#ATC1), not considered in the analysis of the PDTC/ATC group, was run in each experiment together with the tissues, and used as a reference control. Chemiluminescence was detected using the Chemi-Doc-IT Imaging System (UVP). Band intensities were analyzed with the image analysis program NIH ImageJ and the NOX4 signal was normalized to a β-actin signal. Normalized band intensities of the samples were divided for that obtained in the loading control.

### 2.7. GPx Activity

GPx activity was determined in the cytoplasmic fraction of thyroid tumors and contralateral normal tissues by means of a commercial kit (Cayman Chemicals, Ann Arbor, MI, USA). In particular, GPx activity was measured indirectly by a coupled reaction with glutathione reductase, which resulted in the oxidation of NADPH to NADP+ with a reduction of absorbance at 340 nm, and expressed as U per milligram of protein. Each sample was analyzed in duplicate and results were expressed as averages.

### 2.8. Statistical Analyses

A *t*-test (or Wilcoxon non-parametric test) was applied to evaluate relations between continuous variables, using GraphPad Prism software. Correlations were calculated using a Pearson correlation (or Spearman non-parametric correlation). Statistical significance were defined as *p* < 0.05. Gene expression data were log2-transformed and quantile-normalized with R software.

## 3. Results

### 3.1. OS in Thyroid Tissues

In all tumors and benign lesions, H_2_O_2_ generation was significantly higher than that found in the normal tissues analyzed (*p* < 0.0001) (Figure 1, panel A). Among benign lesions, H_2_O_2_ production in HAs was significantly higher compared with FAs (*p* = 0.01). In malignant lesions, FTCs and ATC/PDTCs had an increased H_2_O_2_ production compared with PTCs (*p* = 0.0009 and 0.009, respectively). Moreover, ROS generation in FTCs was higher than that obtained in FAs (*p* = 0.005) and NIFTPs (*p* = 0.01) (Figure 1, panel A). By performing a paired analysis between the benign/malignant tumors and the corresponding normal tissues, a significant difference in H_2_O_2_ production was found for PTCs and FTCs (*p* < 0.0001, 0.003, respectively), and for adenomas (*p* = 0.006), whereas, likely due to the limited number of cases, the trend was not statistically significant for NIFTPs (Figure 1, panel B). This analysis could not be performed for ATC/PDTCs, since the corresponding normal tissues were available only for two samples. 

### 3.2. Correlation between OS and Genetic Profile

Mutational profiles of the tissues analyzed are reported in Table 2. BRAFV600E mutations were found in twelve PTCs and in one PDTC. RET/PTC fusions were found in two PTCs and RAS mutations in three PTCs, in four FTCs, in one NIFTP, in one FA, and 1one HA. TP53 mutations were found in two ATCs and in two PDTCs, one of which also harbored a PTEN duplication. Finally, TERT promoter mutations were found in association with BRAFV600E in five PTCs and in one PDTC, in association with RAS mutations in two FTCs, and in association with a TP53 alteration in one ATC. Correlations between the genetic profile and the H_2_O_2_ levels were calculated only for PTCs, since the number of the other benign or malignant cases were too limited for a reliable statistical analysis. Although, as reported in Figure 1, PTCs taken as a whole showed a significantly higher H_2_O_2_ generation compared with normal tissues, differences were greater in mutated than in non-mutated tumors (BRAF + TERT, BRAF only, BRAF all, RET or RAS and all mutated cases taken together *p* < 0.0001, non-mutated cases taken together *p* = 0.03) (Figure 2, panel A). Consistently, either PTCs harboring BRAF + TERT, or a BRAF mutation (either alone or in combination with TERT, ALL BRAF) or PTCs harboring any mutation (ALL MUT) showed significantly increased ROS production compared with non-mutated PTCs (*p* = 0.03, 0.01, 0.02, respectively).

When we performed a paired analysis with the corresponding normal tissues, a highly significant difference in H_2_O_2_ production was found in BRAF + TERT (*p* = 0.009), BRAF only (*p* = 0.003), ALL BRAF (*p* < 0.0001), RET or RAS mutated (*p* = 0.02), and ALL MUT tumors (*p* < 0.0001) but not in non-mutated PTCs (*p* = 0.12) (Figure 2, panel B).

### 3.3. Correlation between OS with TDS and Clinical and Prognostic Features of Patients

In malignant tumors taken as a whole, TDS was inversely correlated with H_2_O_2_ generation (*p* = 0.03) (Figure 3). To note, TDS in BRAF mutated PTCs was significantly lower than that found in BRAF non-mutated cases (*p* = 0.0001).

We then correlated the OS level with several clinical and prognostic features of differentiated thyroid cancer (DTC) cases (23 PTCs and 6 FTCs, reported in Table 1). Interestingly, the H_2_O_2_ production was found to significantly correlate with tumor stage (*p* = 0.03) and ATA risk (*p* = 0.003), being higher in stages > 1 and in high/intermediate ATA risk cases. Moreover, H_2_O_2_ generation in DTCs showed a trend toward a correlation with extrathyroidal extension (*p* = 0.06) (Figure 3).

### 3.4. NOX4 Protein Expression in Tissues

NOX4 protein expression detected in PTCs, NIFTPs, FAs/HAs, FTCs and ATC/PDTCs was significantly higher compared with that found in normal thyroid tissues (*p* = 0.003, *p* = 0.02, *p* = 0.009, *p* = 0.006, *p* = 0.01, respectively) (Figure 4).

### 3.5. GPx Activity

GPx activity was significantly increased in PTCs ALL and in PTCs ALL MUT (*p* < 0.0001), PTCs NON MUT (*p* = 0.01), NIFTPs, FAs/HAs, FTCs (*p* = 0.003) and PDTCs/ATCs (*p* = 0.0002) compared with normal tissues. No differences in GPx activity were found between different tumor lesions. However, though not significant, GPx activity in mutated PTCs (ALL MUT) was higher than that found in non-mutated tumors (NON MUT) (Figure 5, panel A). Interestingly, GPx activity was inversely correlated to H_2_O_2_ generation considering all benign and malignant lesions (*p* = 0.005), but not in normal tissues (Figure 5, panel B and C). No correlations between GPx activity and clinical and prognostic features of the patients with PTC were found.

## 4. Discussion

Significantly increased oxidative stress (OS) was found in all benign thyroid lesions and malignant thyroid cancers analyzed, compared with the corresponding normal tissues. Among malignant lesions, we detected significantly higher OS in FTC and PDTC/ATC compared with PTC, likely suggesting a correlation of ROS levels with the aggressiveness of the tumors. Consistently, in DTCs a positive correlation was found between ROS generation, ATA risk and stage, indicating that OS might correlate with the worst tumor presentation.

An original result was the significantly higher H_2_O_2_ production in FTCs compared with adenomas, indicating that the multistep progression from benign to malignant forms involves an increase in H_2_O_2_ production, too. Moreover, this finding, if confirmed in the material obtained from fine-needle aspirates, could be exploited for the presurgical differential diagnosis of cytologically indeterminate nodules.

OS is a biochemical state that is due to an imbalance between oxidant and antioxidant systems, and it has a predicted role in thyroid tumorigenesis [7,8]. In thyrocytes, NOX4 is the most abundant NOX isoform and constitutively generates ROS in different intracellular compartments. An increased NOX4 protein expression was found in the cytoplasmic fraction of all neoplastic lesions analyzed compared with normal thyroid tissues, particularly in PTCs and FTCs, in accordance with previous findings [12].

GPx is considered the main antioxidant enzyme in the thyroid gland [19], and its activity has been found to be stimulated by both H_2_O_2_ [17] and NOX4 through the PIM-1 oncogene [30] (mechanisms summarized in Figure 6). Consistently, and in accordance with previous findings [13,20], we found a higher GPx activity in most tumors analyzed compared with normal tissues, with a significant positive correlation with H_2_O_2_ generation, likely indicating an attempt of compensation of antioxidant defenses to the increased production of peroxides. These compensatory antioxidant responses might be necessary to maintain ROS in concentrations allowing cell survival. Nevertheless, the chronic exposure to OS in TC is predicted to contribute to oxidative DNA damage, including base lesions, and single-strand breaks, possibly leading to BRAF and RAS driver mutations [7]. Moreover, exposure of human thyroid cells to H_2_O_2_ is able to induce DNA double-strand breaks that can promote genetic instability, leading for example to oncogenic RET/PTC rearrangement [7]. Accordingly, we found significantly increased OS in mutated PTCs (including BRAF and RAS mutations and RET fusions) compared with non-mutated PTCs. As in a vicious circle, BRAF and RAS oncogenes were found to be able to upregulate NOX4 expression [6,31], and NOX4-derived ROS to mediate RAS-induced DNA damage and senescence and BRAFV600E-induced NIS repression. Consistently, NOX4 expression inversely correlated with TDS in TCGA tumors [6]. These findings, which suggested that genes involved in thyroid differentiation might be silenced by a mechanism controlled by NOX4-derived ROS (Figure 6), are strengthened by our results showing a significant correlation between NOX4-derived ROS and TDS in all malignant thyroid tumors for the first time.

Interestingly, it has been recently shown that NOX4 inhibition increases apoptosis in Lenvatinib-resistant BCPAP cells and xenografts [32]. These findings indicated NOX4 or NOX4-derived ROS as potential therapeutic targets in resistant PTCs.

A drawback of the present study is the relatively limited number of aggressive thyroid cancers analyzed, due to the requirement of processing fresh thyroid tissues for an accurate OS and GPx activity detection and to the rarity of these forms. Moreover, though in the thyroid gland NOX and GPx represent the main ROS-generation and detoxification systems, respectively, we cannot exclude differences in tumor and normal tissues of other pro- or anti-oxidant components.

## 5. Conclusions

In conclusion, the present study gives more insights into the scant knowledge on ROS in thyroid benign and malignant tissues, mostly deriving from Prof. Dupuy and our groups [12,13]. Our data indicate that thyroid tumors are exposed to high OS compared with normal tissues, while showing a compensative increased GPx activity. Moreover, the level of ROS generation correlates with tumor aggressiveness and with the presence of mutations in the MEK–ERK signaling pathway in PTC. The OS status in thyroid tumors warrants attention since it may have diagnostic, prognostic and therapeutic relevance.

## Figures and Tables

**Figure 1 cancers-14-05857-f001:**
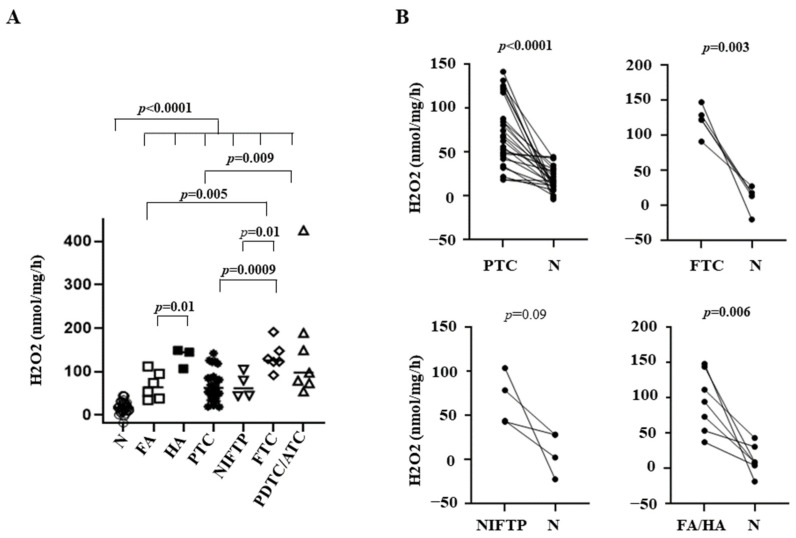
(**A**) H_2_O_2_ generation in thyroid tissues. The median value was plotted; (**B**) Paired analysis of H_2_O_2_ generation in tumors and corresponding normal tissues. N = normal contralateral tissue; FA = follicular adenoma; HA = Hurtle cell adenoma; PTC = papillary thyroid carcinoma; NIFTP = non-invasive follicular thyroid neoplasms with papillary-like nuclear features; FTC = follicular thyroid carcinoma; PDTC/ATC = poorly differentiated thyroid cancer/anaplastic thyroid cancer.

**Figure 2 cancers-14-05857-f002:**
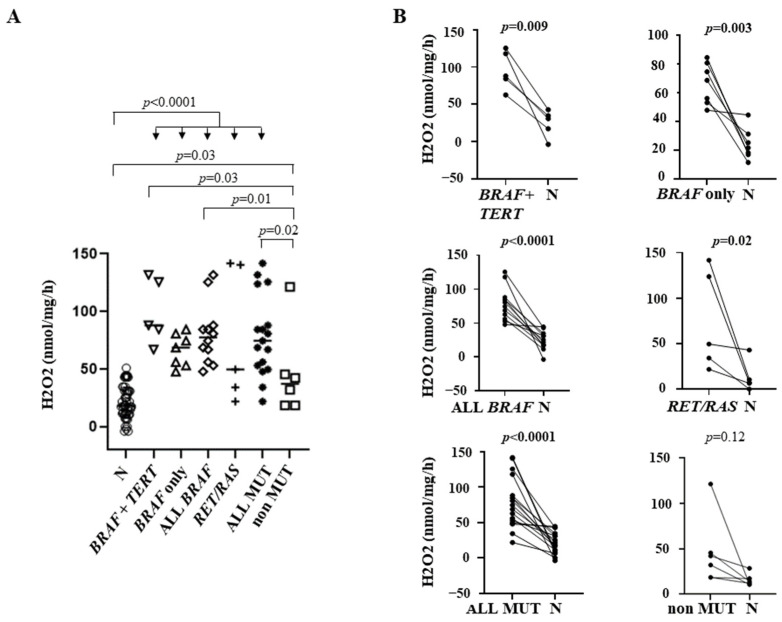
(**A**) H_2_O_2_ generation in papillary thyroid cancers (PTCs) and contralateral tissues, according to the mutational status. The median value was plotted. (**B**) Paired analysis of H_2_O_2_ generation in tumors and corresponding normal tissues. N = normal contralateral tissue; BRAF + TERT = PTC with double BRAF and TERT mutation; BRAF only = PTC with only a BRAF mutation; ALL BRAF = PTC with BRAF mutation alone or in combination; RET/RAS = PTC with RET fusions or RAS mutations; ALL MUT = PTCs with any mutation; non MUT = PTCs without mutations detected (see Methods section).

**Figure 3 cancers-14-05857-f003:**
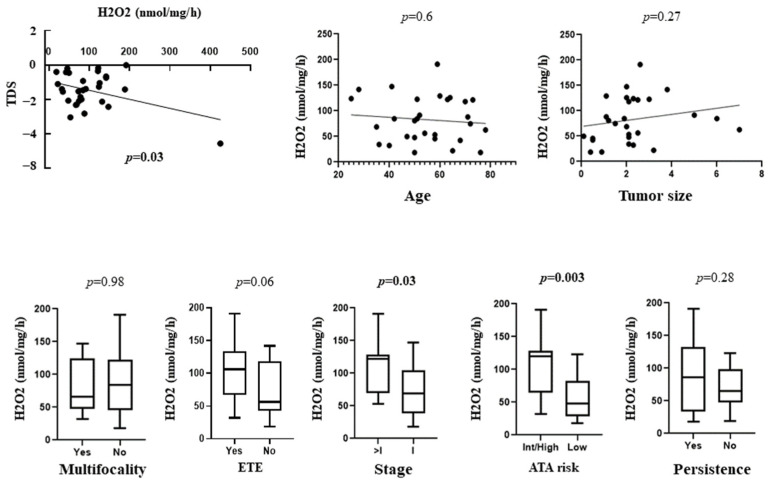
Correlation between H_2_O_2_ generation levels and TDS and clinical and prognostic features of the patients with differentiated thyroid cancer. TDS = thyroid differentiation score; ETE = extrathyroidal extension; ATA = American thyroid association.

**Figure 4 cancers-14-05857-f004:**
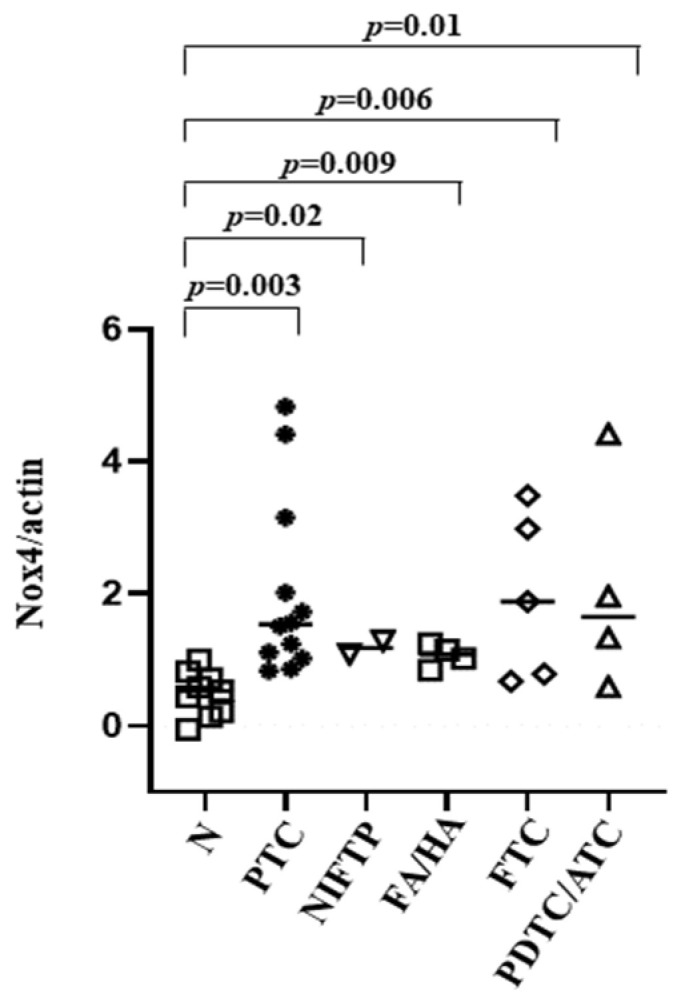
Normalized NOX4/β-actin expression in thyroid tissues. N = normal contralateral tissue; PTC = papillary thyroid carcinoma; NIFTP = non-invasive follicular thyroid neoplasms with papillary-like nuclear features; FA/HA = follicular adenoma/Hurtle adenoma; FTC = follicular thyroid carcinoma; PDTC/ATC = poorly differentiated thyroid cancer/anaplastic thyroid cancer.

**Figure 5 cancers-14-05857-f005:**
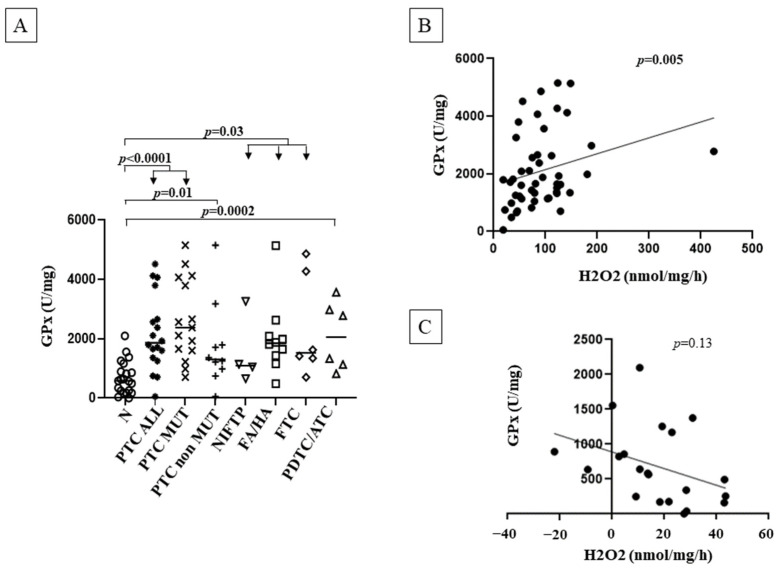
(**A**) GPx activity in thyroid tissues; (**B**) correlation between GPx activity and H_2_O_2_ generation in benign nodules and malignant thyroid tumors; (**C**) correlation between GPx activity and H_2_O_2_ generation in normal contralateral tissues. N = normal contralateral tissue; PTC = papillary thyroid cancer; PTC ALL = all papillary thyroid carcinoma cases; PTC ALL MUT = PTCs with any mutation; PTC non MUT = PTCs without mutations detected (see Methods section); NIFTP = non-invasive follicular thyroid neoplasms with papillary-like nuclear features; FA = follicular adenoma; HA = Hurtle cell adenoma; FTC = follicular thyroid carcinoma; PDTC/ATC = poorly differentiated thyroid cancer/anaplastic thyroid cancer.

**Figure 6 cancers-14-05857-f006:**
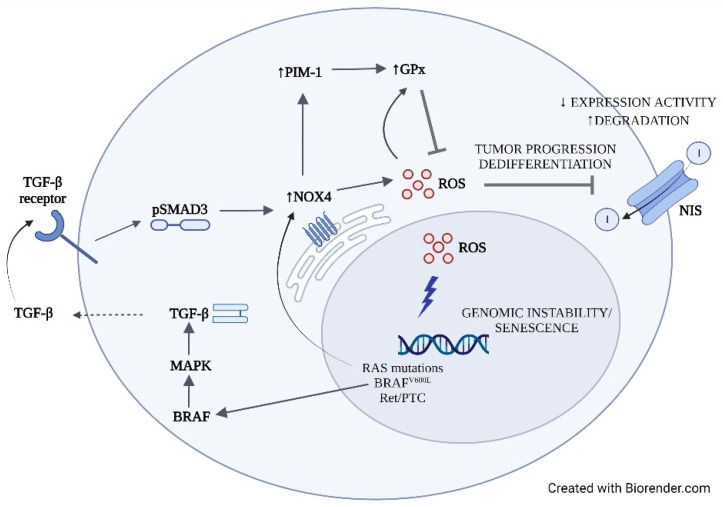
Schematic representation of the cellular mechanisms of reactive oxygen species (ROS) generation and effects.

**Table 1 cancers-14-05857-t001:** Clinical-prognostic features of the patients.

Sample	Age at Diagnosis (Years)	Size (cm)	Multifocality	ETE	ATA Risk	Stage	Persistence
PTC1	64	2	Yes	Yes	Intermediate	II	Yes
PTC2	51	6	No	No	Low	I	No
PTC3	71	1.1	No	Yes	High	II	Yes
PTC4	72	1.5	No	Yes	Low	I	No
PTC5	50	1.2	No	No	Low	I	No
PTC6	50	2.1	No	No	Low	I	No
PTC7	58	2.1	Yes	No	Intermediate	II	No
PTC8	42	1.9	No	No	Intermediate	I	Yes
PTC9	54	2.5	No	No	Low	I	No
PTC10	35	2	Yes	Yes	Intermediate	I	No
PTC11	36	2.1	No	No	Low	I	Yes
PTC12	25	2.3	Yes	Yes	Intermediate	I	Yes
PTC13	47	0.1	Yes	No	Intermediate	I	Yes
PTC14	65	3.2	No	No	Low	I	Yes
PTC15	28	3.8	No	No	Intermediate	I	Yes
PTC16	58	0.5	No	No	Low	I	No
PTC17	68	0.5	Yes	No	Low	I	No
PTC18	76	0.9	No	No	Low	I	No
PTC19	73	2.5	No	No	Intermediate	II	No
PTC20	50	0.4	No	No	Low	I	Yes
PTC21	40	2.3	Yes	Yes	Intermediate	I	Yes
PTC22	70	2.1	No	No	Intermediate	I	No
PTC23	78	7	Yes	Yes	High	II	Yes
FTC1	52	5	No	No	Low	I	No
FTC2	51	3	No	No	Low	I	No
FTC3	41	2	Yes	Yes	High	I	Yes
FTC4	63	3	No	No	Int	II	Yes (deceased)
FTC5	60	1.1	No	Yes	High	Ivb	Yes
FTC6	59	2.6	No	Yes	High	IVa	Yes (deceased)
ATC1	76	4.5	Yes	Yes	High	IVb	Yes (deceased)
ATC2	60	5.5	Yes	Yes	High	IVb	Yes (deceased)
ATC3	65	4	Yes	Yes	High	IVb	Yes (deceased)
PDTC1	74	3.6	Yes	Yes	High	IVb	Yes (deceased)
PDTC2	42	3.5	Yes	Yes	High	IVb	Yes (deceased)
PDTC3	80	2	Yes	Yes	Int	II	Yes (deceased)
PDTC4	34	3.5	Yes	Yes	High	II	Yes

ETE: extrathyroidal extension; ATA risk according to Haugen et al., Thyroid 2016 [25]; PTC: papillary thyroid cancer; FTC: follicular thyroid cancer; ATC: anaplastic thyroid cancer; PDTC: poorly differentiated thyroid cancer.

**Table 2 cancers-14-05857-t002:** Mutational profiles of thyroid tumors.

Sample	Mutation
PTC1	*BRAF* p.V600E + *TERT*-124C > T
PTC2	*BRAF* p.V600E + *TERT*-124C > T
PTC3	*BRAF* p.V600E + *TERT*-124C > T
PTC4	*BRAF* p.V600E + *TERT*-124C > T
PTC5	*BRAF* p.V600E + *TERT*-124C > T
PTC6	*BRAF* p.V600E
PTC7	*BRAF* p.V600E
PTC8	*BRAF* p.V600E
PTC9	*BRAF* p.V600E
PTC10	*BRAF* p.V600E
PTC11	*BRAF* p.V600E
PTC12	*BRAF* p.V600E
PTC13	*RET*/PTC1
PTC14	*RET*/PTC3
PTC15	*NRAS* p.Q61K
PTC16	*NRAS* p.Q61R
PTC17	*NRAS* p.Q61R
PTC18	WT
PTC19	WT
PTC20	WT
PTC21	WT
PTC22	WT
PTC23	WT
NIFTP1	WT
NIFTP2	WT
NIFTP3	*NRAS* p.Q61R
NIFTP4	WT
FA1	*NRAS* p.Q61R
FA2	WT
FA3	WT
FA4	WT
FA5	WT
FA6	WT
FA7	WT
HA1	WT
HA2	*NRAS* p.Q61R
HA3	WT
FTC1	*NRAS* p.Q61R + *TERT*-124C > T
FTC2	*NRAS* p.Q61R
FTC3	*NRAS* p.Q61R
FTC4	WT
FTC5	*NRAS* p.Q61R + *TERT*-124C > T
FTC6	WT
ATC1	*TP53* c.722_724delCCT + *TERT*-124C > T
ATC2	WT
ATC3	*TP53* p.R175H
PDTC1	*TP53* c.866_867delTC
PDTC2	WT
PDTC3	*BRAF* p.V600E + *TERT*-124C > T
PDTC4	*TP53* p.C135Y, p.M133R, c.920-2A > G+ *PTEN* c.741dupA

## Data Availability

RNA-Sequencing data were deposited in the Harvard Dataverse (https://dataverse.harvard.edu/, accessed on 21 October 2022) with accession number: https://doi.org/10.7910/DVN/B6CRS2, accessed on 21 October 2022.

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
