# Peer review of "Oxidative Stress Correlates with More Aggressive Features in Thyroid Cancer"

_cancers, 2022, doi:10.3390/cancers14235857_

Round 1
Reviewer 1 Report
The impact of oxidative stress (OS) in the pathogenesis and progression of thyroid cancer has been already reported and it is not a new concept. Aim of the study was to investigate OS in malignant, benign thyroid tumors and normal tissues and additionaly to correlate of OS with the genetic and expression profile of the tumors and clinico-pathologic features of the patients. Authors state that OS was inversely correlated with tumor differentiation and directly correlated with the presence of somatic mutations and higher tumor aggresiveness. The study is interesting, well constructed and paper is well writin. An important limitation of current study was low numer of aggresive thyroid cancers included to the study The study gives more insights into the knowledges on OS in thyroid cancer and encourages to further studies to assess it diagnostic, prognostic and therapeutic relevance. I rcommend this paper for publication
Author Response
We thank the Reviewer for her/his appreciation of our study
Reviewer 2 Report
Muzza et al. present sound analysis of oxidative stress parameters in various neoplastic thyroid tissues and corelate the results with mutational status and molecular phenotype.
The results are highly original and may have practical clinical significance. However, I noticed a minor spelling error that needs to be corrected. Please see p. 2, line 51 - there is "..differentiate thyroid cancer...", it should be "..differentiated thyroid cancer..."
Author Response
The results are highly original and may have practical clinical significance.
We thank the Reviewer for her/his appreciation of our study.
However, I noticed a minor spelling error that needs to be corrected. Please see p. 2, line 51 - there is "..differentiate thyroid cancer...", it should be "..differentiated thyroid cancer..."
The spelling error was corrected in the revised version of the paper.